# Low-dose self-amplifying mRNA COVID-19 vaccine drives strong protective immunity in non-human primates against SARS-CoV-2 infection

Amy R. Rappaport[1,3], Sue-Jean Hong[1,3], Ciaran D. Scallan[1,3], Leonid Gitlin[1], Arvin Akoopie[1], Gregory R. Boucher[1], Milana Egorova [1], J. Aaron Espinosa [1], Mario Fidanza[1], Melissa A. Kachura[1], Annie Shen[1], Gloria Sivko[2], Anne Van Abbema[1], Robert L. Veres [1] & Karin Jooss[1✉]

The coronavirus disease 2019 (COVID-19) pandemic continues to spread globally, highlighting the urgent need for safe and effective vaccines that could be rapidly mobilized to immunize large populations. We report the preclinical development of a self-amplifying mRNA (SAM) vaccine encoding a prefusion stabilized severe acute respiratory syndrome coronavirus 2 (SARS-CoV-2) spike glycoprotein and demonstrate strong cellular and humoral immune responses at low doses in mice and rhesus macaques. The homologous prime-boost vaccination regimen of SAM at 3, 10 and 30 μg induced potent neutralizing antibody (nAb) titers in rhesus macaques following two SAM vaccinations at all dose levels, with the 10 μg dose generating geometric mean titers (GMT) 48-fold greater than the GMT of a panel of SARS-CoV-2 convalescent human sera. Spike-specific T cell responses were observed with all tested vaccine regimens. SAM vaccination provided protective efficacy against SARS-CoV-2 challenge as both a homologous prime-boost and as a single boost following ChAd prime, demonstrating reduction of viral replication in both the upper and lower airways. The SAM vaccine is currently being evaluated in clinical trials as both a homologous prime-boost regimen at low doses and as a boost following heterologous prime.

[1] Gritstone bio, Inc., Emeryville, CA, USA. [2] Battelle Biomedical Research Center, West Jefferson, OH, USA. [3] These authors contributed equally: Amy R. Rappaport, Sue-Jean Hong, Ciaran D. Scallan. ✉email: kjooss@gritstone.com

The current severe acute respiratory syndrome coronavirus 2 (SARS-CoV-2) pandemic has spurred the rapid development and approval of multiple vaccines; however, the virus continues to spread globally with devasting humanitarian and economic tolls. Novel vaccine platforms, such as self-amplifying mRNA (SAM), which could potentially be dose sparing due to the platform's ability to replicate post vaccination, should be explored to address the ongoing or future pandemics.

Two main vaccine platforms have come to the fore in this pandemic, mRNA (mRNA-1273, Moderna; BNT162b2, Pfizer/BioNTech) and adenovirus based vaccines (ChAdOx1/AZD1222, AstraZeneca; Ad26.COV2.S, Janssen), both of which have demonstrated protection from hospitalization and death[1–4]. Adenovirus (Ad) based vaccine vectors have previously been shown to be potent inducers of both humoral and cellular immunity, and the authorized adenovirus COVID vaccines demonstrate effectiveness in the range of 64.3–71%[2,5]. mRNA-based vaccines are relatively new to the clinic, with the BNT162b2 and mRNA-1273 COVID vaccines being the first to receive emergency use authorization. Both have demonstrated potency in preclinical and clinical testing[1,6,7], with a standard vaccination regimen of two doses given 3–4-weeks apart. The authorized mRNA COVID vaccine doses are either 30 μg for the BNT162b2 vaccine or 100 μg for mRNA-1273, and both provide clinical protection in the range of 88–93%[5]. Production of mRNA vaccines by a cell-free transcription reaction enables simple downstream purification and rapid vaccine release, providing for a cost-effective manufacturing process. Non-viral delivery systems, such as lipid nanoparticles (LNPs), allow repeated RNA administrations without inducing anti-vector immunity.

SAM vaccines offer the potential benefit of driving potent immune responses at low doses, as the mRNA replicates intracellularly, leading to high and durable antigen expression[8]. We have developed and tested SAM based vaccines, initially in immuno-oncology applications, and have demonstrated their safety and potency in boosting neoantigen-specific T-cell responses primed by a chimpanzee adenovirus (ChAd68) vaccine in clinical trials (manuscript in press).

In this study, we evaluated a SAM-SARS-CoV-2 vaccine either as a homologous prime/boost vaccine regimen (SAM/SAM), or as a boost vaccine following a ChAd-SARS-CoV-2 prime (ChAd/SAM) in both mice and non-human primates (NHP). We demonstrate that SAM in both homologous and heterologous regimens can provide protection against SARS-CoV-2 replication post infection in NHP, comparable to currently authorized conventional mRNA vaccines and achieved at lower doses.

## Results

SAM and ChAd vaccines encoding full-length SARS-CoV-2 spike were evaluated for immunogenicity in mice. Serum was collected at various timepoints post a single immunization and assessed for total anti-S1 IgG and pseudovirus neutralizing antibodies (nAb). Due to the different kinetics of the humoral and cellular immune response, a separate cohort of mice was immunized and splenocytes collected at 2-weeks post-immunization to assess T-cell response by IFNγ ELISpot. While the ChAd-Spike(V1) vaccine induced detectable nAb titers and spike-specific T-cell response (Fig. 1B, C), similar to those observed with other adenoviral spike vectors[9], the immune response induced by the SAM-Spike(V1) vaccine was significantly higher (Fig. 1B, D). This had not been observed before with other antigens, prompting optimization of the spike sequence to improve expression. Highly variable expression was observed with various codon-optimized spike sequences (Fig. 1A) in the context of ChAd and increased expression in vitro correlated with higher nAb titers in mice

(up to 63-fold) with a trend toward increased T-cell response (up to 1.7-fold) with the V2 sequence compared to the V1 sequence (Fig. 1B, C). A SAM-Spike(V2) vaccine resulted in similar immune response to that induced by the ChAd-Spike(V2) vaccine (Fig. 1B–D). Both ChAd & SAM Spike(V2) vaccines demonstrated increased durability of spike IgG titers compared to Spike(V1) (Supplementary Fig. 1). The Spike(V8) variant generated a similar immune response to Spike(V2) (Supplementary Fig. 1C, D).

To further improve vaccine potency, vectors expressing the prefusion-stabilized spike protein[10] were generated. The Spike(V2) sequence was modified by mutation of the furin cleavage site (FurinΔ) and introduction of proline (P) substitutions to the S2 domain (2P: K986P, V987P) (Supplementary Fig. 2A), which have been shown to stabilize the spike protein in its prefusion form, leading to increased immunogenicity[11]. The unmodified Spike(V2) is mostly in its uncleaved format with two smaller species at approximately 100 and 75 kDa. Incorporation of the furin mutation results in the loss of the 100 kDa band with the 75 kDa band still present and most likely reflects the TMPRSS2 cleaved product[12]. The 2P and 6P substitutions appear to accentuate cleavage at the furin site as manifested by an intense band at 100 kDa but not at the TMPRSS2 site and the combination of FurinΔ and the 2P and 6P mutations maintains the Spike protein in its uncleaved form (Supplementary Fig. 2B). The F2P variant demonstrated a further increase in nAb titers compared with V2 for both ChAd (2.9-fold) and SAM (2.1-fold) (Fig. 1E, Supplementary Fig. 2G). Vaccines with additional P-substitutions also increased nAb titers compared with V2 (Supplementary Fig. 2D, E) while T-cell responses remained potent but unchanged by the modifications (Supplementary Fig. 2C, F). Optimized Spike(V2)-F2P ChAd and SAM vaccines were evaluated in mice, as either a homologous (SAM/SAM) or heterologous (ChAd/SAM) regimen. Both regimens generated broad spike-specific T-cell responses to multiple epitopes spanning the spike antigen following prime immunization, which were increased following SAM boost (2.5 for heterologous and 5.2-fold for homologous) and were predominantly CD8+ and $T_H$1-biased (Fig. 2A; Supplementary Fig. 3 and 4). Spike-specific nAb titers were also increased by SAM boost (36-fold for heterologous and 22-fold for homologous) (Fig. 2B, Supplementary Fig. 3B). This demonstrates the increased immunogenicity of both heterologous and homologous prime/boost regimens utilizing ChAd and SAM vaccines compared to single dose regimens, similar to what has previously been observed in mice[13].

The robust immune response induced by SAM in mice, even at low doses (Supplementary Fig. 5), was explored further in NHP. Groups of 5 rhesus macaques received two immunizations of SAM-Spike(V2)-F2P (4-week interval) at either 3, 10, or 30 μg (Fig. 3A, Supplementary Table 1). Spike S1-specific IgG titers were observed in some NHP following a single SAM immunization in a dose-dependent manner and were increased to high levels in all NHP following a 2nd immunization at all doses (GMT 8880, 13,145, and 1223 at 30, 10, and 3 μg, respectively, Fig. 3B). Spike-specific T-cell responses were detected at all dose levels following two immunizations (Fig. 3C) and were significantly increased compared to control animals for all dose levels (adjusted p-value = 0.012, 0.024, and 0.024 for 30, 10, and 3 μg, respectively, by one-tailed Mann–Whitney test with Bonferroni correction for multiple comparisons). Strong IFNγ and minimal IL-4 responses were observed 1-week post boost in all vaccinated animals, indicating a $T_H$1-biased response (Fig. 3D). Strong nAb titers were detected following two SAM immunizations by both a pseudovirus neutralization (PNA) and live virus microneutralization assay (MNA), which strictly correlated (Fig. 3E, F; Supplementary Fig. 6). Similar nAb titers were detected in all ten NHP at both the 30 and 10 μg dose level (NT50

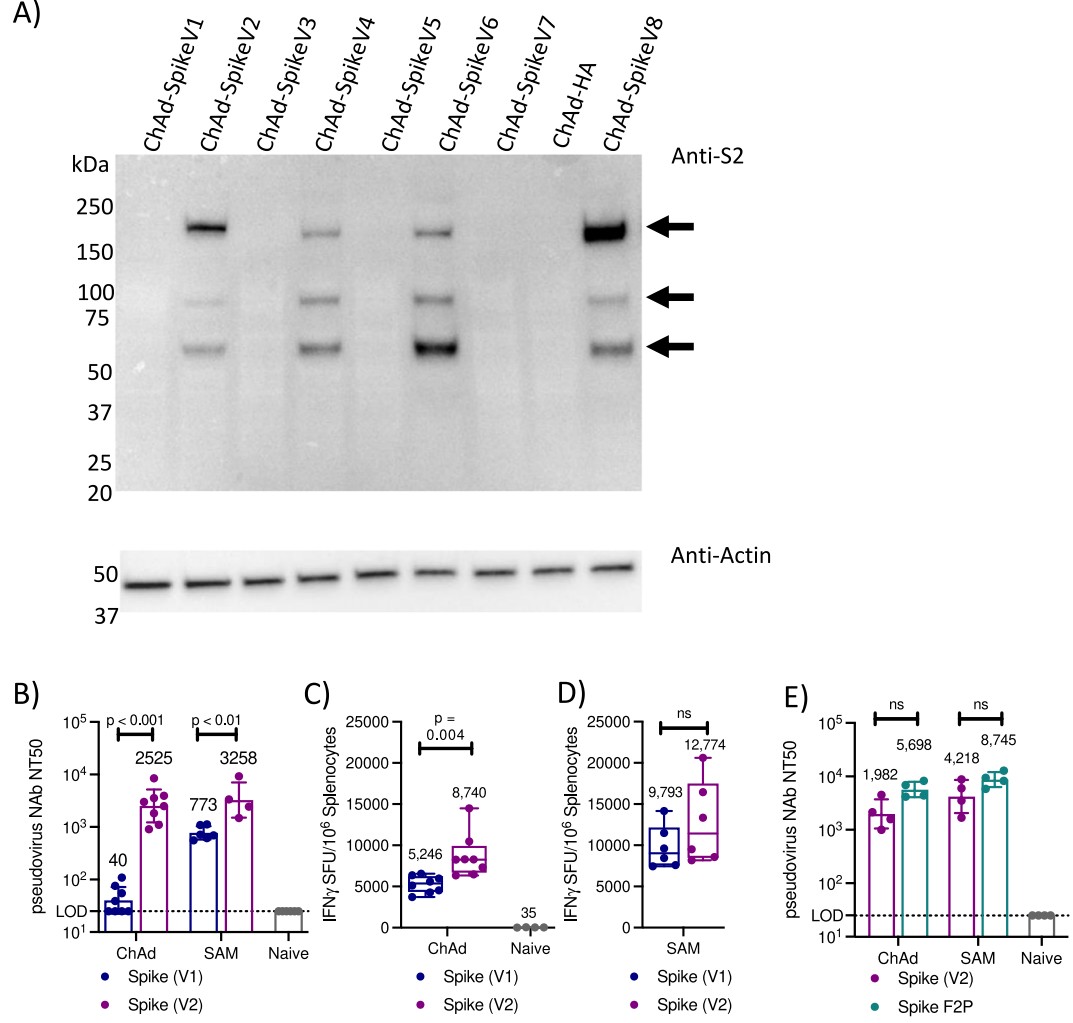

**Fig. 1 Codon optimization of spike sequence increases antigen expression in vitro and immune response in mice. A** Expression analysis of 8 codon-optimized spike ChAd vaccine vectors (V1–V8), assessed by Western blot 40 h post infection in HEK293F cells using an anti-spike S2 antibody. A ChAd-influenza HA construct served as a negative control. Arrows indicate full-length spike and S2 cleavage products. Samples were run on a second gel in parallel and an anti-Actin antibody was used to confirm equivalent protein loading for all samples. Representative data from one of 3 repeat experiments. **B** Serum pseudovirus nAb titers in female Balb/c mice 8 weeks post single immunization with either ChAd ($1 \times 10^{11}$ VP) or SAM (10 μg) encoding either the spike V1 ($n = 8$ for ChAd, $n = 6$ for SAM) or V2 sequence ($n = 8$ for ChAd, $n = 4$ for SAM). Mice were immunized and sera collected separately for each vaccine, samples were assessed by validated pseudovirus neutralization assay. Naive samples all below LOD (LOD = 25). Geometric mean (annotated) and geometric SD, two-tailed Mann–Whitney test ($p = 0.000155$ for ChAd and $p = 0.009524$ for SAM). **C** Spike-specific T-cell response assessed by IFNγ ELISpot (sum of eight pools) in Balb/c mice. Splenocytes collected 2 weeks post immunization with ChAd ($1 \times 10^{11}$ VP) encoding either the spike V1 or V2 sequence ($n = 8$/group). **D** Spike-specific T-cell response assessed by IFNγ ELISpot (sum of two pools) in Balb/c mice. Splenocytes collected 2 weeks post immunization with SAM (10 μg) encoding either the spike V1 or V2 sequence ($n = 8$/group). **C**, **D** Box and whiskers represent median, IQR and range. Mean annotated, $p$-value determined by two-tailed unpaired Student's $t$-test. Note that data in (**C**) and (**D**) are from separate studies. **E** Pseudovirus nAb titers in mouse sera 8 weeks post immunization with ChAd ($1 \times 10^{11}$ VP) or SAM (10 μg) encoding the specified spike variant. Geomean (annotated) and geometric SD. All mice immunized and assessed together, representative data from one of three studies. Sera from Naive mice were below LOD of 25. Two-tailed Mann–Whitney tests.

GMT 2180 and 2590 2-weeks post boost by MNA). Lowering the dose of SAM to 3 μg resulted in overall lower, but detectable nAb titers in 5/5 NHP (GMT 245 2-weeks post boost by MNA) (Fig. 3F). Potent spike-specific nAb titers, higher than those previously reported to offer protection from SARS-CoV-2 infection in NHP[14], were observed at all dose levels. While potent immune responses were detected with all dose levels, there was no increase in serum IFNα levels following SAM immunizations at doses ≤10 μg, which was clearly detectable at the 30 μg dose in this study, suggesting that innate immune pathways are not strongly activated at lower SAM doses (Supplementary Fig. 7).

One month following the 2nd SAM immunization (Fig. 3A), NHP were challenged with SARS-CoV-2. Consistent with previous studies of SARS-CoV-2 challenge in rhesus macaques, none of the vaccinated or control animals showed clinical signs of illness[6,7,14–17]. To assess vaccine efficacy in controlling SARS-CoV-2 replication post infection, viral load was assessed by PCR in bronchial alveolar lavage (BAL) fluid, oropharyngeal swabs, and nasal swabs post challenge, similar to other studies of SARS-CoV-2 vaccines in NHP[7,9,15]. In addition to total genomic RNA, subgenomic mRNA (sgRNA) was assessed to measure virus replication[18]. Three days post-challenge, 1/5 animals in the 30 μg and 3 μg groups and 0/5 animals in the 10 μg group had

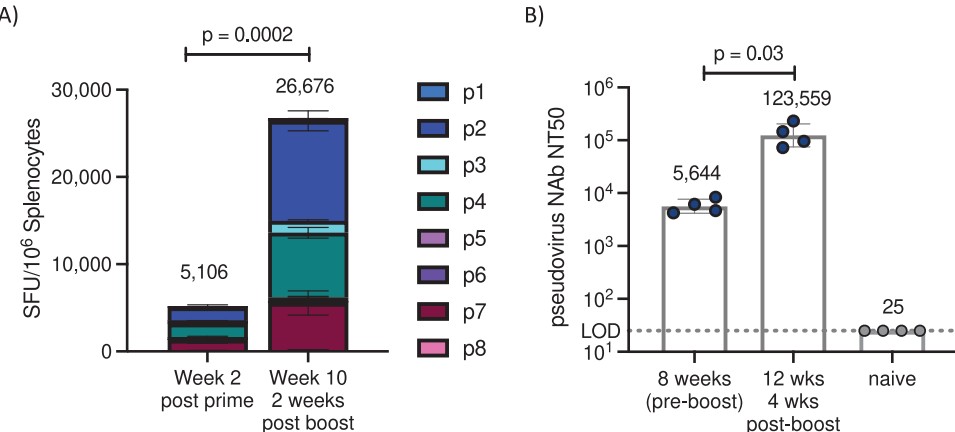

**Fig. 2 SAM boost immunization drives increased antigen-specific T cells and neutralizing antibody titers.** Mice received SAM Spike(V2)-F2P prime immunization, followed by boost 8 weeks post prime (10 μg each). $N = 4$ mice/group. **A** T-cell response at the specified timepoint, assessed by IFNγ ELISpot following overnight stimulation with eight peptide pools spanning spike antigen. Mean ± SEM for each peptide pool for $N = 4$ independent animals per group; representative data from one of four similar studies, see additional data in Supplementary Fig. 3. Annotated numbers are the average sum of all eight peptide pools for each animal. Unpaired Student's t-test (two-tailed) comparing sum of pools for each animal at each timepoint. Immunizations for each group were staggered to enable splenocyte collection and measurement of multiple timepoints post immunization in the same assay. **B** Pseudovirus neutralizing titer (NT50) at the specified timepoint. Geometric mean and geometric SD of $N = 4$ mice per group for one experiment. Representative data from one of three similar studies. Sera from Naive mice were below LOD of 25. Two-tailed Mann–Whitney test.

detectable sgRNA in BAL, as compared to 5/5 of the controls, demonstrating reduced viral replication at all dose levels (Fig. 4A). Comparison of peak viral titer for each animal between day 1 and day 10 post challenge demonstrated significant protection from viral replication at the 30 μg dose level compared to control animals (adjusted $p$-value = 0.021, Kruskal–Wallis followed by Dunn's multiple comparison post-test, Supplementary Fig. 8). On day 3, none of the animals in the SAM 10 μg dose group and 2/5 in the SAM 30 and 3 μg groups had detectable sgRNA in nasal swabs, compared to 4/5 in the control group, demonstrating significantly decreased viral replication in nasal swabs at the 10 μg dose level ($p$ = 0.026, peak viral load between day 2 and 14 post challenge compared to control, Kruskal–Wallis followed by Dunn's multiple comparison post-test, Fig. 4C and Supplementary Fig. 8). Furthermore, total RNA levels were lower in BAL, oropharyngeal and nasal swabs in all SAM vaccinated animals compared to control animals, with improved viral clearance observed with the 30 and 10 μg of SAM compared with 3 μg SAM (Supplementary Fig. 9). This data demonstrates protective immunity in all SAM vaccinated animals, even at the 3 μg dose.

The potency of the SAM vaccine was further explored in a ChAd/SAM prime/boost regimen ($5 \times 10^{11}$ VP ChAd prime with 30 μg SAM boost, 6-week interval, Fig. 3A). Spike-specific IgG titers were observed following a single ChAd immunization, as reported for other adenovirus vaccines[15,17] and were increased 2.5-fold following SAM boost to similar titers as those observed following two SAM immunizations (10 or 30 μg) (Fig. 3B). Higher spike-specific $T_H1$ biased T-cell responses were detected after a single ChAd immunization compared to a single SAM immunization (at any dose), demonstrating the potency of the ChAd vaccine to rapidly drive high T-cell titers (Fig. 3C, D). T-cell responses were increased following the SAM boost to similar levels as those observed following two SAM immunizations. Spike-specific nAb titers were detected in 10/10 NHP following a single ChAd immunization (MNA NT50 GMT 663 4-weeks post prime in ChAd only treatment group) and were increased 2.3-fold following SAM boost (Fig. 3E, F). NHP were challenged with SARS-CoV-2 six weeks following a single ChAd immunization and at 4-weeks following the SAM boost (Fig. 3A).

On day 3 post challenge, only 1/5 animals in the ChAd only and 0/5 animals in the ChAd/SAM groups had detectable sgRNA in BAL, as compared to all control animals. On day 3, 1/5 NHP in the heterologous prime/boost group had detectable sgRNA in the nasal swab, compared to 4/5 in the ChAd only and control groups, demonstrating increased protective efficacy with heterologous prime/boost compared to a single ChAd immunization (adjusted $p$-values = 0.034 with ChAd/SAM and $p > 0.99$ with ChAd only, Kruskal–Wallis with Dunn's multiple comparison post-test, peak viral load between day 2 and 10 post challenge compared to control, Fig. 4C and Supplementary Fig. 8). Collectively, these data demonstrate protection from SARS-CoV-2 replication post infection provided by the vaccines, with the 30 and 10 μg of SAM homologous prime/boost and the heterologous regimen providing complete protection from viral replication post infection and the 3 μg SAM homologous prime/boost offering slightly decreased immune response but clearly providing protection to animals challenged with live virus.

## Discussion

Following extensive antigen sequence optimization, we demonstrate that a SAM vaccine encoding SARS-CoV-2 spike induces humoral and cellular immune responses in mice and NHP and provides protection from SARS-CoV-2 replication post-infection at low doses. A SAM-SARS-CoV-2 spike vaccine expressing full-length spike induced strong antigen-specific T-cell responses at low doses (Supplementary Fig. 5) and high nAb responses following a single immunization in mice (Fig. 1B, E). Unexpectedly, the ChAd vaccine expressing the first-generation spike antigen induced significantly weaker immune responses compared to SAM, prompting extensive sequence optimization, which resulted in selection of a spike sequence that strongly increased antibody and T-cell titers (Fig. 1). The difference in immunogenicity likely reflects differences in the biology of the two vaccine platforms, such as RNA splicing and routing to the cytosol required for ChAd but not SAM.

We evaluated the SAM vaccine in NHP to study immune correlates of protection. Spike-specific nAb titers were observed following two immunizations of SAM at low doses ranging from 3 to 30 μg. Potent nAb titers were generated at the 10 μg dose of

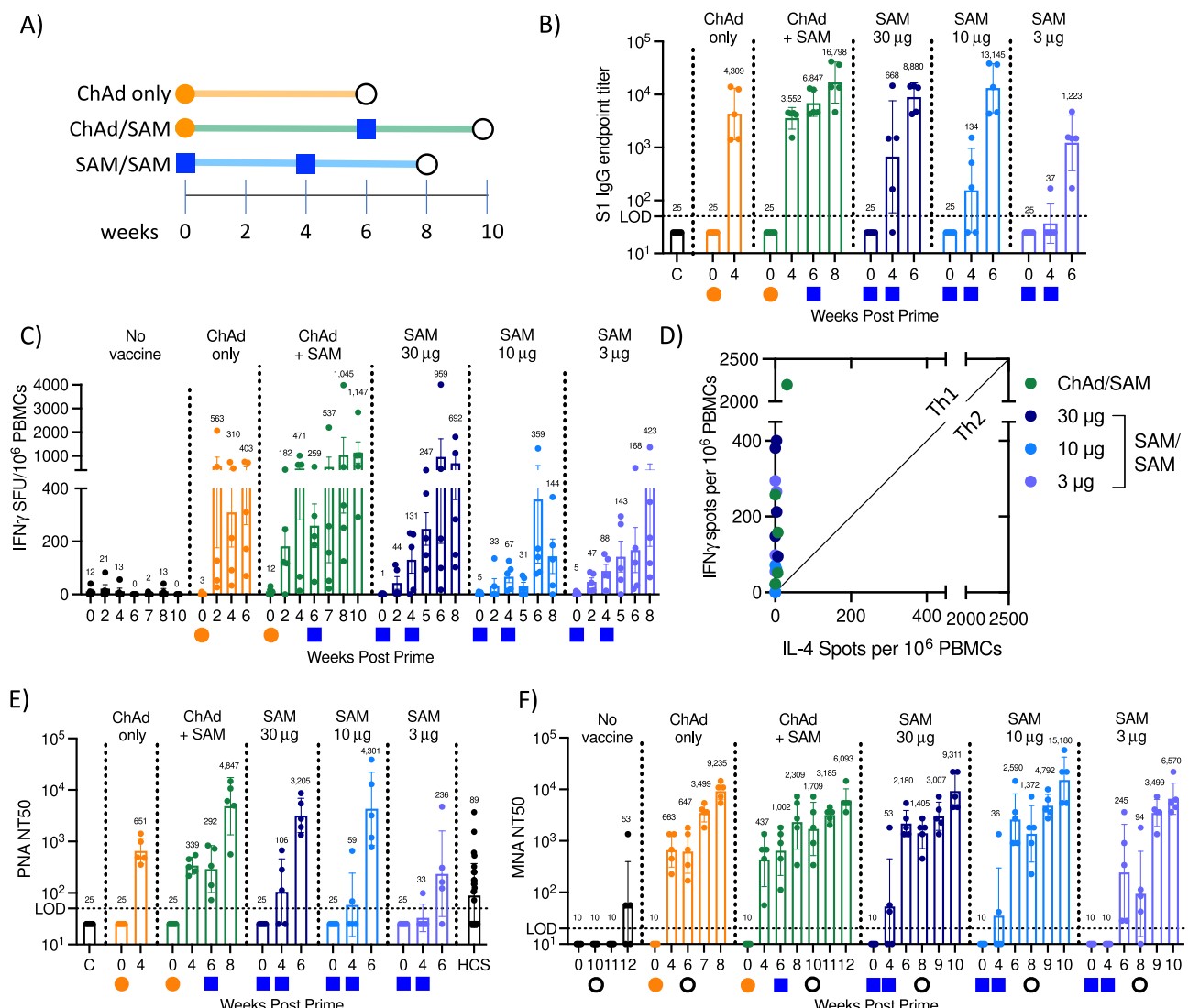

**Fig. 3 Immunogenicity in rhesus macaques. A** Schematic of SARS-CoV-2 vaccine evaluation and challenge in rhesus macaques. Orange circles represent ChAd immunization ($5 \times 10^{11}$ VP), blue squares represent SAM vaccination at varying doses and black circle represents challenge with SARS-CoV-2 virus. N = 5/group. **B** Spike S1 IgG endpoint titers, geometric mean (annotated), and geometric SD. LOD = 50, samples below LOD set to ½ LOD. **C** Spike-specific T-cell responses assessed by overnight IFNγ ELISpot at specified timepoint post immunization. Mean (annotated) ± SEM. **D** IL-4 vs IFNγ ELISpot of PBMCs assessed 1-week post boost immunization. Individual animal values (n = 5/group). Diagonal line represents unity line. Positive control stimulated wells were too numerous to count accurately. **E** Pseudovirus neutralization titers assessed in sera at specified timepoint post immunization. LOD = 50, samples <LOD set to ½ LOD. Geometric mean (annotated) and geometric SD. HCS = human convalescent serum assessed by same assay. C = PBS control injected animals assessed 8 weeks post injection. **F** Live virus micro-neutralizing titer at specified timepoint post immunization and post SARS-CoV-2 challenge, geometric mean (annotated), and geometric SD. LOD = 20, samples below LOD set to ½ LOD.

SAM, even after a single immunization, which increased following a 2nd immunization to levels similar to that observed with 10-fold higher doses (100 µg) of authorized SARS-CoV-2 mRNA vaccines[6,7,16] in NHP, suggesting that SAM might present a dose sparing vaccine option. This may be due to the replication of SAM and increased mRNA half-life, which may lead to stronger and potentially more durable immune responses at lower doses compared to non-replicating mRNA vaccines. Another COVID-19 SAM vaccine, COVAC1, currently in clinical development, has been shown to induce overall weaker immune responses than the authorized mRNA vaccines, with only a 61% seroconversion rate at the 10 µg dose[19], compared to 100% with either the Pfizer/BioNTech mRNA vaccine at doses ranging from 10–30 µg[20] or the Moderna mRNA vaccine at 50 and 100 µg[21]. This decreased immunogenicity may have resulted from activation of innate immune pathways that can partly restrict SAM replication and

antigen expression. Another SAM vaccine platform that uses an alternative non-prefusion stabilized spike sequence and a different LNP formulation, ARCT-021, induced low nAb titers below the GMT titer of convalescent sera, demonstrating decreased immunogenicity compared to the authorized mRNA vaccines[22]. Notably, our SAM vaccine differs from the COVAC1 and ARCT-021 SAM platforms in the LNP, the spike sequence and the SAM backbone. During the preclinical development of our SAM SARS-CoV-2 vaccine, we demonstrated that each of these components impact vaccine potency.

One difference between the authorized mRNA and the SAM vaccines in development is the use of modified nucleosides in the mRNA vaccines, which have been shown to reduce innate immunity post vaccination and may lead to increased vaccine-induced adaptive immune response. Induction of innate immunity and its impact on vaccine potency might differ between

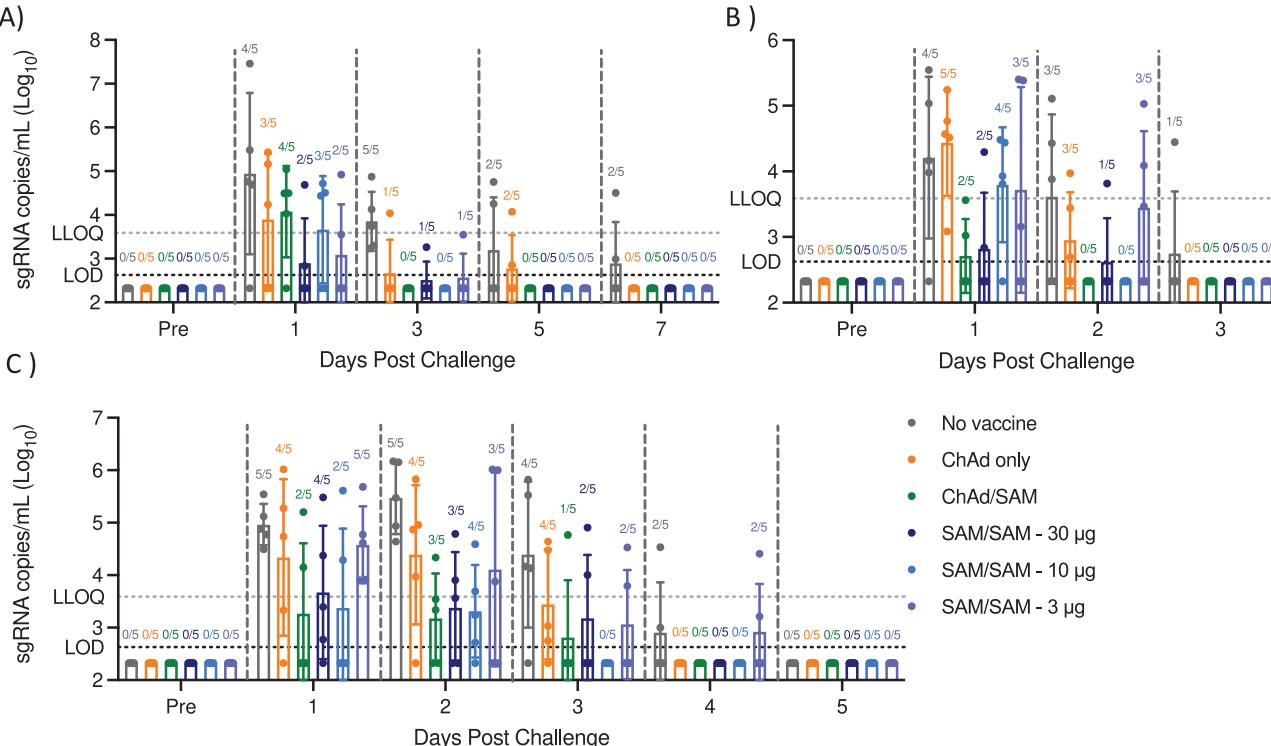

**Fig. 4 Viral replication is reduced in all vaccinated NHP following SARS-CoV-2 challenge.** Subgenomic RNA levels determined by RT-qPCR at specified timepoint post SARS-CoV-2 challenge for each animal in **A** bronchial alveolar lavage, **B** oropharyngeal swab, or **C** nasal swab. LOD = 422, samples below LOD set to ½ LOD. Geometric mean and SD. Numbers above each bar are numbers of NHP with viral load levels >LOD. Additional timepoints post complete clearance of viral RNA from vaccinated animals are not shown for clarity, full dataset provided in Supplementary tables.

mRNA and SAM vaccines, due to the replication of the SAM vaccine, which is an area of intense investigation. During the development of our SAM vaccine, various vaccine components were optimized to decrease activation of innate immune pathways. The NHP data suggest that our SAM SARS-CoV-2 vaccine does not induce strong systemic innate immunity at doses ≤10 μg (Supplementary Fig. 7).

Spike-specific $T_H1$ biased T-cell responses were observed in SAM immunized NHP at all doses, although with high heterogeneity across animals, potentially due to the small group size, lack of randomizing animals based on their MAMU haplotype, and/or variability in age and gender (Supplementary Table). Accumulating evidence has demonstrated the importance of T-cells in clearance of virus-infected cells and reduction of disease severity[14,23,24]. T-cell responses are associated with improved survival in B-cell deficient COVID-19[+] patients with hematologic cancer, who mount weak humoral responses to the authorized COVID vaccines[25]. Vaccine-induced T-cells may also contribute to broader protection against SARS-CoV-2 variants of concern that are sub-optimally neutralized by spike antibodies[26,27]. Furthermore, T-cell memory may be more durable than B-cell memory and thus induction of both arms of the immune system may provide longer protection against future coronavirus variants[28].

We have demonstrated that homologous SAM prime/boost drives humoral and cellular immune responses and provides protection from SARS-CoV-2 replication post-infection at low doses in NHP. The SAM vaccine will be evaluated in a Coalition for Epidemic Preparedness Innovations (CEPI) sponsored clinical trial in South Africa. We further demonstrate that SAM provides a potent boost following a ChAd prime. The heterologous prime/boost concept has been evaluated clinically using Ad and mRNA vaccines, which led to comparable or increased antibody responses, compared to homologous boosts[29,30]. While a single vaccine platform is preferred for simplicity of manufacturing and rollout during a pandemic, a heterologous Ad prime and SAM boost regimen may drive faster, stronger, and more consistent T-cell responses, and therefore, may offer an attractive option to individuals who are immune compromised, such as the elderly or patients on chemotherapy. We have shown that our ChAd/SAM prime/boost regimen drives robust T-cell responses in patients who received concurrent chemotherapy[31] (Clinical trials NCT03639714, NCT03953235, manuscript in review). The heterologous prime/boost concept is currently being evaluated in clinical trials, utilizing our SARS-CoV-2 SAM vaccine as a boost in individuals who have been primed with adenoviral vaccines (NCT04776317).

In summary, SAM is a potent vaccine platform that drives robust T-cell and nAb responses at low doses, protects NHP from SARS-CoV-2 replication post-infection and is currently being tested in humans in both homologous and heterologous prime/boost regimens. It is adaptable, rapid to manufacture, and offers an additional vaccine platform in the fight against the ongoing SARS-CoV-2 pandemic and current and emerging infectious pathogens, as well as cancer.

## Methods

**Codon optimization**. Spike nucleotide sequences based on the original Wuhan-Hu-1 strain (MN908947: https://www.ncbi.nlm.nih.gov/nuccore/MN908947.3) with the D614G mutation were designed for maximum expression in Homo Sapiens using three different codon optimization tools. Spike-V1 was generated using the IDT (Coralville IA) optimization tool, V2-V7 were codon optimized using the COOL optimization algorithm (version 20160707a)[32] and V8 was optimized using an SGI-DNA (La Jolla, CA) codon optimization tool (version 20190802). Each codon-optimized sequence was ordered as double-stranded DNA gBlocks (IDT) with at least a 30 bp overlap with sequences flanking the ChAd68 cloning site.

**Vector generation**. The ChAd68 nucleotide sequence was based on the wild-type sequence obtained by MiSeq (Ilumina sequencing) of virus obtained from the ATCC (VR-594). The sequence of a E1 (578–3404 bp)/E3 deleted virus (2125–31,825 bp) was assembled into pUC19 from VR-594-derived and synthetic (SGI-DNA) fragments. An E4 deletion between E4ORF2-4 was introduced by PCR. A CMV promoter/enhancer with an SV40 polyA was introduced into the E1 region and the spike gBlock sequences introduced by Gibson assembly (Codexis) and transformed into Stbl4 (Thermo Fisher) cells. Error-free clones were selected by PCR and sequencing and plasmid DNA prepared at the Maxi-prep scale (Machery-Nagel). Furin and proline spike mutations, 2P or 6P, were introduced into the spike protein by overlapping PCR extension using primers to introduce the specific mutations.

The pA68-E4d-Spike plasmids were linearized, purified using a Nucleospin kit (Machery-Nagel), and transfected into 2 mL of HEK293F cells (0.5 mL/mL) using TransIT-Lenti (Mirus Bio). The virus was amplified, harvested, and re-infected into 30 mL of HEK293F cells for 48–72 h. Cells and media were harvested and used to infect 400 mL of HEK293F cells. Cells were harvested after 48 h and lysed by a freeze/thaw step (−80 °C/37 °C) in 10 mM Tris pH 8.0/0.1% Triton-X100, and then purified by two successive rounds of CsCl gradient centrifugation. Virus bands were purified and dialyzed 3x into 1X ARM buffer (10 mM Tris pH 8.0, 25 mM NaCl, 2.5% glycerol). Viral particle concentration was determined by the Absorbance 260 nm method post lysis in 0.1% SDS and the infectious unit (IU) titer was determined by an immunostaining assay that uses an anti-adenovirus capsid antibody[33]. In brief, limiting dilutions of the virus are added to HEK293A cells and incubated for 48 h at 37 °C/8% CO$_2$. Cells are then fixed in ice-cold methanol for a minimum of 20 min, washed with 1xPBS, and then blocked for 1 h with 1% BSA in 1xPBS. A rabbit anti-adenovirus antibody diluted 1:8,000 in 1% BSA/PBS was added to the cells for 1 h. The cells were washed with PBS, 4 x and then a HRP conjugated anti-mouse antibody (Bethyl Labs) was added for 1 h before washing. The signal was detected using DAB, 3,3′ Diaminobenzidine tetrahydrochloride (Sigma-Aldrich), colorimetric substrate, and stained cells were counted. The total IU/mL was then calculated by calculating the number of stained cells per field of view (based on an average of 10 fields/replicate), the number of fields of view per well, the dilution used, and the volume of virus added per well.

Spike sequences were PCR amplified and cloned into PacI/BstBI sites of a pUC02-VEE vector. Capped SAM was synthesized in vitro using HiScribe T7 Quick High Yield RNA Synthesis Kit (New England Biolabs) and purified using a RNeasy Maxi Kit (Qiagen) according to the manufacturer's protocol. SAM was subsequently encapsulated in a lipid nanoparticle (LNP) using a self-assembly process in which an aqueous solution of SAM is rapidly mixed with a lipid mixture in ethanol[34]. RNA encapsulation efficiency was measured using Ribogreen RNA quantitation reagent (Thermo Fisher) and confirmed to be >95% in all batches analyzed. SAM-LNP was formulated into a buffer containing 5 mM Tris (pH 8.0), 10% sucrose, 10% maltose.

**Western analysis**. HEK293F cells (Thermo) seeded at 5e5 cells/mL were infected with an MOI of 1 IU/cell. Cells were incubated between 40 and 72 h before harvesting by centrifugation. Cells were washed in 1xPBS and resuspended in 1XSDS/PAGE buffer with 2.5% β-mercaptoethanol, then sheared through a 27 g needle, prior to incubation at >90 °C for 5 min. Samples (10 μL) were separated on a 4–20% SDS-PAGE gel and blotted onto a PVDF membrane using a Trans-Blot Turbo transfer system (BioRad). Membranes were blocked for 1 h in 5% skim milk in TBST (Tris-buffered saline plus 0.1% Tween-20) and then probed with an anti-S2 mouse monoclonal antibody (GeneTex) at a 1:1000 dilution for 2 h to overnight. Membranes were washed (0.05% Tween 20 in 1X TBS) and then probed with a Rabbit anti-Mouse HRP antibody at a 1:10,000 dilution (Bethyl Labs) for 1 h before washing and detection with a SuperSignal West Femto Maximum Sensitivity Substrate (Pierce). A parallel blot using a mouse anti-actin antibody (Bethyl Labs) at a 1:1000 dilution was used to ensure equivalent protein amounts per well.

**Mouse studies**. Mouse studies were conducted at Murigenics (Vallejo, CA) under IACUC-approved protocols. *M. musculus* (mice), Balb/cAnNHsd strain (Envigo), 6–8 weeks old females, were used for all studies. Mice were housed in 19–21 °C and 50 ± 20% relative humidity, in rooms with at least ten room air changes per hour, in ventilated cage racks with micro-filtered tops with Sanichip bedding (#7090A). Photoperiod was diurnal; 12 h light, 12 h dark. Food was irradiated Harlan Teklad ad libitum, sterile water provided ad libitum. All immunizations were bilateral intramuscular to the tibialis anterior, 2 injections of 50 μL each. Mouse spleens were harvested 12–14 days following immunization. Spleens were suspended in RPMI complete (RPMI + 10% FBS) and dissociated using the gentleMACS Dissociator (Milltenyi Biotec). Dissociated cells were filtered using a 40 μm strainer and red blood cells were lysed with ACK lysing buffer (150 mM NH$_4$Cl, 10 mM KHCO$_3$, 0.1 mM EDTA). Following lysis, cells were filtered with a 30 μm strainer and resuspended in RPMI complete. At various timepoints post immunization, 200 μL of blood was drawn. Blood was centrifuged at 1000 × g for 10 min at room temperature. Serum was collected and frozen at −80 °C.

**Intracellular cytokine staining**. Freshly isolated splenocytes were resuspended at a density of 5 × 10$^6$ cells/mL in complete RPMI and following an overnight rest at 4 °C, 1 × 10$^6$ cells per well were distributed into v-bottom 96-well plates. Cells were pelleted and resuspended in 100 μL of complete RPMI containing an overlapping peptide pool containing 316 peptides (each 15 amino acids in length, 11 amino acid overlap) spanning the SARS-CoV-2 spike antigen, at a final concentration of 0.5 μg/mL per peptide (Genscript). A second well with DMSO only was used as a negative control for each sample. After 1 h of incubation at 37 °C, Brefeldin A (Biolegend) was added to a final concentration of 5 μg/mL and cells were incubated for an additional 4 h. Following stimulation, cells were washed with PBS and stained with fixable viability dye (eBioscience, e506). Extracellular staining was performed in FACS buffer (PBS + 2% FBS + 2 mM EDTA). Cells were then washed, fixed, and permeabilized with the eBiosciences Fixation/Permeabilization Solution Kit. Intracellular staining was then performed in permeabilization. Samples were collected on a Cytoflex LX (Beckman Coulter). Analysis of flow cytometry data was performed using FlowJo software (version 10.7.1).

**Antibodies and dilutions**. All antibodies used in this study were commercially available and are as follows with amounts/dilutions provided:

*Flow cytometry*. Rat Anti-Mouse IFNγ clone XMG1.2, Invitrogen, Cat #: 12-7311-82, PE (1 μL/1 × 10$^6$ cells).
Rat Anti-Mouse TNFα clone MP6-XT22, eBioscience, Cat#17-7321-82, APC (1.25 μL/1 × 10$^6$ cells).
Rat Anti-Mouse IL2 clone JES6-5H4, eBioscience, Cat#48-7021-82, eFluor450 (1.25 μL/1 × 10$^6$ cells).
Rat Anti-Mouse IL4 clone 11B11, BioLegend, Cat#504132, PE-Dazzle594 (1.25 μL/1 × 10$^6$ cells).
Rat Anti-Mouse IL10 clone JES5-16E3, eBioscience, Cat#56-7101-82, AF700 (1.25 μL/1 × 10$^6$ cells).
Rat Anti-Mouse CD4 clone GK1.5, BioLegend, Cat#100451, BV605 (2.5 μL/1 × 10$^6$ cells).
Rat Anti-Mouse CD8 clone 53-6.7, BD, Cat#563332, BV786 (0.75 μL/1 × 10$^6$ cells).
Live/dead stain, eBioscience Cat#65-0866-18, eFluor506 (1:1000).

*ELISpot*. Anti-monkey IFNγ, clone 7-B6-1, biotin-conjugated (1:1000), Mabtech, Kit catalog #3421M-4APW-10.
Anti-mouse IFNγ, clone R4-6A2, biotin-conjugated (1:1000), Mabtech, Kit catalog #3321-4APW-10.
Anti-human IL4, clone IL4-II, biotin-conjugated (1:1000), Mabtech, Kit catalog #3410-4APW-10.

*IU Assay*. Rabbit anti-Adenovirus antibody, Abcam Cat# Ab6982 (1:8000).
Goat anti-Rabbit HRP, Bethyl Labs, Cat#A120-101P (1:1000).

*Western blot*. Mouse anti-Spike S2 monoclonal antibody clone 1A9, GeneTex, Cat#GTX632604 (1:1000).
Rabbit Anti-Mouse Actin polyclonal, Bethyl Labs, Cat#A300-485A (1:1000).
Goat Anti-Mouse HRP, Bethyl Labs, Cat#A90-116P (1:10,000).
Goat anti-Rabbit HRP, Bethyl Labs, Cat#A120-101P (1:10,000).

Commercial antibodies were validated by the vendor.
Commercial antibody information and validation info where applicable:
https://www.thermofisher.com/antibody/product/IFN-gamma-Antibody-clone-XMG1-2-Monoclonal/12-7311-82
https://www.thermofisher.com/antibody/product/TNF-alpha-Antibody-clone-MP6-XT22-Monoclonal/17-7321-82
https://www.thermofisher.com/antibody/product/IL-2-Antibody-clone-JES6-5H4-Monoclonal/48-7021-82
https://www.biolegend.com/nl-nl/products/pe-dazzle-594-anti-mouse-il-4-antibody-10715
https://www.thermofisher.com/antibody/product/IL-10-Antibody-clone-JES5-16E3-Monoclonal/56-7101-82
https://www.biolegend.com/en-us/products/brilliant-violet-605-anti-mouse-cd4-antibody-10708?GroupID=BLG4745
https://www.bdbiosciences.com/en-us/products/reagents/flow-cytometry-reagents/research-reagents/single-color-antibodies-ruo/buv805-rat-anti-mouse-cd8a.612898 https://www.thermofisher.com/order/catalog/product/65-0866-14
https://www.genetex.com/Product/Detail/SARS-CoV-SARS-CoV-2-COVID-19-spike-antibody-1A9/GTX632604 https://www.thermofisher.com/antibody/product/Cytoskeletal-Actin-Antibody-Polyclonal/A300-485A
https://www.abcam.com/adenovirus-type-5-antibody-ab6982.html

**Eukaryotic cell lines**. HEK293F (Thermo, #11625019).
HEK293-ES (Expression Systems, LLC, #94-007F).
HEK293A (Thermo, R70507).
Vero-E6 (ATCC, CRL-1586).
Vero-E6 (BEI, NR-596).
Cell lines were authenticated by original vendors. All cell lines tested negative for mycoplasma. No commonly misidentified lines were used.

**Non-human primate studies**. Study was conducted in compliance with all relevant local, state, and federal regulations and was approved by the Battelle Institutional Animal Care and Use Committee (IACUC), Protocol #B06082. Thirty Indian-origin male and female rhesus macaques (*M. mulatta*) >2.5 years old (Envigo) were housed at Battelle (Columbus, Ohio). NHP were vaccinated with either ChAd-Spike(V2)-F2P (Group 1—$5 \times 10^{11}$ VP, study day 0; Group 2—$5 \times 10^{11}$ VP, study day 28), SAM-Spike(V2)-F2P (Group 1—30 µg, study day 42; Group 3—30 µg, study days 14 and 42; Group 4—10 µg, study days 14 and 42; Group 5—3 µg, study days 14 and 42), or PBS (Group 6, study days 0 and 42). All injections were bilateral intramuscular, 0.5 mL per leg (1 mL total) to the thigh. Note that immunizations were staggered to enable challenge immunization on study day 70 for all groups. Each group was divided into two sub-groups and study initiated 1 week apart, to enable two staggered challenge groups. Animals were challenged with SARS-CoV-2 (strain USA-WA1/2020) via the intratracheal (0.5 mL) and intranasal (0.25 mL per nostril) routes with a target dose of ~$1.6 \times 10^6$ PFU (undiluted). Serum was collected using serum separator tubes and whole blood collected in cell preparation tubes and peripheral blood mononuclear cells isolated and cryopreserved in CryoStorCS solution (Sigma-Aldrich).

**ELISpot assays**. IFNγ and IL-4 ELISpot assays were performed using pre-coated 96-well plates (Mabtech, Monkey IFNγ ELISpot PLUS, ALP; Mouse IFNγ ELISpot PLUS, ALP; or Human IL-4 ELISpot PLUS, ALP) following manufacturer's protocol. For NHP, frozen PBMCs were thawed at 37 °C and then rested overnight in RPMI + 10% FBS. $1 \times 10^5$ PBMCs were plated per well in triplicate with a single overlapping peptide pool spanning spike from the N to C terminus (GenScript, 15 amino acid length, 11 amino acid overlap, 314 peptides total) at final concentration of 1 µg/mL per peptide and incubated overnight at 37 °C in RPMI + 10% FBS. For mouse studies, freshly isolated splenocytes were stimulated overnight with either two (~120 peptides/each) or eight different overlapping peptide pools (36–40 peptides each) spanning the SARS-CoV-2 spike antigen, at a final concentration of 1 µg/mL per peptide (GenScript). Splenocytes were plated in duplicate at $1 \times 10^5$ cells per well and $2.5 \times 10^4$ cells per well (mixed with $7.5 \times 10^4$ naive cells) for each stimulus. DMSO only was used as a negative control for each sample. For both mouse and NHP IFNγ ELISpot, positive control wells stimulated with Phorbol Myristate Acetate (PMA, 15 ng/mL, Invivogen) and Ionomycin (500 ng/mL, Invivogen) were included on each plate in triplicate. For NHP IL-4 ELISpot, positive control wells stimulated with Staphylococcal enterotoxin B (SEB, 1000 ng/mL, Sigma-Aldrich) were included on each plate in triplicate. Note that for all assays, positive control wells were too numerous to count accurately (well confluence >35%). For all ELISpot assays, triplicate wells with media only were also included on each plate. Plates were washed with PBS and then incubated with anti-monkey or anti-mouse IFNγ mAb biotin (Mabtech) for 2 h, followed by an additional wash and incubation with Streptavidin-ALP (Mabtech) for 1 h. After final wash, plates were incubated for 10 min with BCIP/NBT (Mabtech) to develop the immunospots. Wells were imaged and spots enumerated using AID reader with vSpot v7 (Autoimmun Diagnostika). Samples with replicate well variability (variability = variance/(median + 1) >10 and median >10 were excluded[35]. Spot values were adjusted based on the well saturation according to the formula: AdjustedSpots = RawSpots + 2*(RawSpots*Saturation/(100−Saturation)[36]. Each sample was background corrected by subtracting the average value of the negative control wells. Data were normalized to spot forming units (SFU) per $1 \times 10^6$ cells by multiplying the corrected spot number by $1 \times 10^6$/cell number plated. Data processing was performed using the R programming language and graphed using GraphPad Prism 9.

**Pseudovirus neutralization assay**. Mouse and human convalescent serum samples (courtesy of Helen Chu, University of Washington) were assessed by Nexelis (Laval, Quebec). NHP serum samples were assessed by Gritstone bio (Emeryville, CA) using the same pseudovirus, controls, reagents, and protocol. Pseudotyped virus particles were made using a genetically modified Vesicular Stomatitis Virus from which the glycoprotein G was removed (VSVΔG). The VSVΔG virus was transduced in HEK293-ES cells (Expression Systems, LLC) previously transfected with the spike glycoprotein of the SARS-CoV-2 coronavirus (Wuhan strain) for which the last 19 amino acids of the cytoplasmic tail were removed (ΔCT). The generated pseudovirus particles (VSVΔG – Spike ΔCT) contain a luciferase reporter which can be quantified in relative luminescence units (RLU). Heat-inactivated serum samples were serially diluted (7-serial 2-fold dilution) in a 96-well plate and a pre-determined amount of pseudotyped virus (corresponding to between approximately 75,000 and 300,000 RLU/well) was applied to the plate and incubated with serum/plasma to allow binding of the neutralization antibodies to the pseudotyped virus.

After the incubation of the serum/plasma-pseudotyped virus complex, the serum/plasma-pseudotyped virus complex was transferred to the plate containing Vero-E6 cells (ATCC). Test plates were incubated at 37 °C with 5% $CO_2$ overnight. Luciferase substrate was added to the plates which were then read using a plate reader detecting luminescence. The intensity of the light being emitted is inversely proportional to the amount of anti-SARS-CoV-2 neutralizing Spike antibodies bound to the VSVΔG – Spike ΔCT particles.

Each microplate was read using a luminescence microplate reader with SoftMax Pro software v6.5.1 (SpectraMax). The dilution of serum required to achieve 50% neutralization (NT50) when compared to a non-neutralized pseudoparticle control was calculated for each sample dilution and the NT50 is interpolated from a linear regression using the two dilutions flanking the 50% neutralization. Data analysis was performed using the R programming language and graphed using GraphPad Prism 9.

**Microneutralization assay**. The microneutralization assay was performed at Battelle Memorial Institute (Columbus, Ohio) to assess the neutralizing antibody titer in serum samples collected from NHPs following vaccination and post-challenge. The virus neutralization titer is expressed as the reciprocal value of the highest dilution of the serum which still inhibited virus replication. All serum samples were analyzed in duplicate.

Briefly, 2-fold serial dilutions of heat-inactivated serum sample were pre-incubated with the virus for 60 min at 37 °C. The virus/serum mixture was then added to a 90 to 100% confluent monolayer of Vero-E6 cells (BEI, Cat. No. NR-596) in 96-well plates and incubated for two days at 37 °C with 5% $CO_2$. Following incubation, the inoculum was removed, and monolayers were incubated for 30 min in 80% cold acetone to allow cell fixation. Plates were incubated with anti-nucleocapsid protein primary antibody cocktail (clones HM1056 and HM1057) (EastCoast Bio, North Berwick, ME) for 60 min at 37 °C (Battelle Memorial Institute, Patent Number 63/041,551 Pending, 2020). The plates were washed and the secondary antibody (goat anti-mouse IgG Horse Radish Peroxidase (HRP) conjugate; Fitzgerald, North Acton, MA) was added to the wells, and the plates were incubated for 60 min at 37 °C[1]. After the plates were washed, the substrate was added, and the plates were incubated at 37 °C. Stop solution was added, and the plates were read for optical density at 405 nm wavelength. Neutralizing activity is defined as at least 50% reduction in signal from the virus-only (VC) wells relative to cells control (CC) wells following the formula [(average VC – average CC)/2] + average CC. The median neutralizing titer ($MN_{50}$) was calculated using Spearman–Kärber analysis method[37].

**S1 IgG ELISA**. 96-well QuickPlex plates (Meso Scale Discovery, Rockville, MD) were coated with 50 µL of 1 µg/mL SARS-CoV-2 S1 (ACROBiosystems, Newark, DE), diluted in DPBS (Corning, Corning, NY), and incubated at 4 °C overnight. Wells were washed three times with agitation using 250 µL of PBS + 0.05% Tween-20 (Teknova, Hollister, CA) and plates blocked with 150 µL Superblock PBS (Thermo Fisher Scientific, Waltham, MA) for 1 h at room temperature on an orbital shaker. Test sera was diluted at appropriate series in 10% species-matched serum (Innovative Research, Novi, MI) and tested in single wells on each plate. Wells were washed and 50 µL of the diluted samples were added to wells and incubated for 1 h at room temperature on an orbital shaker. Wells were washed and incubated with 25 µL of 1 µg/mL SULFO-TAG labeled anti-species antibody (MSD), diluted in DPBS + 1% BSA (Sigma-Aldrich, St. Louis, MO), for 1 h at room temperature on an orbital shaker. Wells were washed and 150 µL Read Buffer T (MSD) added. Plates were read immediately using the QPlex SQ 120 (MSD) ECL plate reader with Methodical Mind software v1.0.37. For each sample, the endpoint titer was calculated as the reciprocal serum dilution at 2-fold the average background value for the plate, interpolated by a linear regression using the two highest dilutions that gave values >2-fold the average background. Data analysis was performed using the R programming language and graphed using GraphPad Prism 9.

**IFNα2α ELISA**. NHP serum samples were analyzed for levels of IFNα2α using a MSD U-PLEX Biomarker assay (catalog number K15068L-2), according to the manufacturer's instructions. Plates were read immediately using the QPlex SQ 120 (MSD) ECL plate reader with Methodical Mind software. Analyte concentration (pg/mL) was calculated using serial dilutions of known standards. Data analysis performed in Microsoft Excel. Each animal and timepoint was run in technical duplicates.

**Viral RNA RT-qPCR**

*Nucleocapsid protein (N1) genomic analysis*. Briefly, RNA was isolated using the Indispin QIAcube HT Pathogen Kit (Indical Bioscience, Germany) on the QIAcube HT instrument (Qiagen, Germany). The isolated RNA was then evaluated in RT-qPCR using the TaqMan Fast Virus 1-step Master Mix (Thermo Fisher Scientific) on a QuantStudio Flex 6 Real-Time PCR System (Applied Biosystems, Foster City, CA). The primers and probe were specific to the SARS-CoV-2 nucleocapsid gene, corresponding to the N1 sequences from the Centers for Disease Control and Prevention (CDC) 2019-Novel Coronavirus (2019-nCoV) Real-Time RT-PCR Diagnostic Panel (https://www.cdc.gov/coronavirus/2019-ncov/lab/rt-pcr-panel-primer-probes.html) except that the probe quencher was modified to Non-Fluorescent Quencher-Minor Groove Binder (NFQ-MGB) (Thermo Fisher Scientific). A standard curve comprised of synthetic RNA containing the target sequence from SARS-CoV-2 isolate WA1 sequence (GenBank Accession Number MN985325.1) (Bio-Synthesis, Inc., Lewisville, TX) was included on each PCR plate for absolute quantitation of SARS-CoV-2 RNA copies in each sample. Thermo-cycling conditions were as follows: Stage 1—50 °C for 5 min for one cycle; Stage 2—95 °C for 20 s for one cycle; Stage 3—95 °C for 3 s and 60 °C for 30 s for 40 cycles. Data analysis was performed using the QuantStudio 6 software-generated values (total copies per well of each sample) and additional calculations to determine SARS-CoV-2 N1 copies per mL of fluid.

*Envelope protein (E) subgenomic analysis.* Following isolation and evaluation using the N1 genomic assay, the isolated RNA was then evaluated as described above using primers and probes specific to the SARS-CoV-2 E gene based on previously described sequences[18], and the reverse primer and probe sequences previously described[38] (Integrated DNA Technologies, Iowa). Primer/probes used were as follows with final concentrations: N1 forward, 5′-GACCCCAAAATCAGC-GAAAT-3′ (500 nM); N1 reverse, 5′-TCTGGTTACTGCCAGTTGAATCTG-3′ (500 nM); N1 probe, 5′-FAM-ACCCCGCATTACGTTTGGTGGACC-NFQ-MGB (125 nM); Esg forward, 5′-CGATCTCTT GTAGATCTGTTCTC-3′ (400 nM); Esg reverse, 5′-ATATTGCAGCAGTACGCACACA-3′ (400 nM); Esg probe: 5′-FAM-ACACTAGCCATCCTTACTGCGCTTCG-MGB-NFQ-3′ (200 nM). A standard curve comprised of synthetic RNA containing the target sequence from SARS-CoV-2 isolate WA1 sequence (GenBank Accession Number MN985325.1) (Bio-Synthesis, Inc., Lewisville, TX) was included on each PCR plate for absolute quantitation of SARS-CoV-2 copies in each sample. Thermocycling conditions were as follows: Stage 1—50 °C for 5 min for one cycle; Stage 2—95 °C for 20 s for one cycle; Stage 3—95 °C for 3 s and 60 °C for 30 s for 40 cycles. Data analysis was performed using the QuantStudio 6 software-generated values (total copies per well of each sample) and additional calculations to determine SARS-CoV-2 E gene subgenomic (Esg) RNA copies per mL of fluid.

**Statistics**. For evaluating disease progression post-challenge in NHPs sample size was pre-determined using the power procedure in SAS (version 9.4). Sample sizes of 5 per group will provide >81% power to detect a 2.6-fold change between each vaccinated dose group and the control group as well as >80% power to detect a 3.3-fold change between any two vaccinated dose groups for any of the continuous parameters that are log-transformed for the analysis. This assumes a 40% coefficient of variation for the continuous parameter. For parameters that are not log-transformed, the sample sizes will provide >80% power to detect a 4.9 standard-unit change between each vaccinated dose group and the control group as well as >80% power to detect a 6.2 standard-unit change between any two vaccinated dose groups assuming a standard deviation of two. These power calculations performed are for one-sided *t*-tests comparing vaccinated animals to controls and two-sided *t*-tests comparing any two of the vaccinated groups using a Bonferroni multiple comparison adjustment to control the overall Type I error level at no more than 5% for each set of tests.

*P* values reported for mouse ELISpot comparisons determined by unpaired Student's *t*-test. *P* values reported for mouse PNA comparisons determined by two-tailed Mann–Whitney tests.

Comparison of MNA and PNA values assessed by nonlinear regression analysis (least-squares fit).

For NHP T-cell analysis, peak values for each animal over the duration of the study (prior to challenge) were compared for each of three SAM vaccinated groups compared to control group by nonparametric one-tailed Mann–Whitney tests. Reported *p*-values are adjusted using Bonferroni's correction for multiple comparisons.

For NHP RT-qPCR analysis peak sgRNA value for each animal post challenge was compared for each vaccinated group to the control group by nonparametric Kruskal–Wallis ANOVA followed by Dunn's multiple comparisons post-test. Adjusted *p*-values are reported.

All statistical analysis performed using GraphPad Prism 9.

**Reporting summary**. Further information on research design is available in the Nature Research Reporting Summary linked to this article.

## Data availability

The data generated in this study are provided in the Source data file. Source data are provided with this paper.

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

## Acknowledgements

We thank Nexelis for running the pseudovirus neutralization assay on mouse sera and their support in transferring the assay for assessing non-human primate sera. We thank Battelle Memorial Institute for execution of the non-human primate challenge study and performing the live virus microneutralization and RT-PCR assays. Preclinical development and in vivo evaluation in mice were partially funded by a Bill & Melinda Gates Foundation grant. The non-human primate challenge study as well as the performance of the microneutralization and RT-PCR assays were funded with federal funds from the National Institute of Allergy and Infectious Diseases, National Institutes of Health, Department of Health and Human Services, under NIAID's Preclinical Services Contract No. HHSN272201800003I/75N93020F00003.

## Author contributions

C.D.S., L.G., S.J.H., and K.J. designed vaccine constructs. C.D.S., S.J.H., J.A.E., R.L.V., L.G., A.A., and M.F. produced the vaccines and performed in vitro evaluation. A.R.R., A.S., A.V.A., M.E., M.A.K., and G.B. designed and executed mouse studies and performed and analyzed serum IgG and ELISpot assays. A.R.R. and G.S. designed NHP study. G.B. performed and analyzed NHP ELISpot assays and cytokine MSD. M.A.K. performed and analyzed NHP PNA assays and M.E. performed and analyzed NHP IgG assays. G.S. oversaw performance and analysis of MNA and RT-PCR assays. A.R.R., S.J.H., C.D.S., and K.J. wrote the manuscript.

## Competing interests

A.R.R., S.J.H., C.D.S., L.G., A.A., G.R.B., M.E., J.A.E., M.F., M.A.K., A.S., A.V.A., R.L.V., and K.J. are stockholders and either current or previous employees at Gritstone Bio, Inc. and may be listed as co-inventors on various pending patent applications related to the vaccine platform presented in this study. The remaining authors declare no competing interests.
