## [Peer Review File · Nature Communications]

Peer review comments first round -

Reviewer #1 (Remarks to the Author):

Rappaport and colleagues developed and tested a self-amplifying mRNA COVID-19 vaccine in mice and NHPs. The topic is interesting, the results are nice. The manuscript is not clearly written, important information is missing, and, in general, it is hard to find the technical/experimental details in the paper. I indicated some of these issues.

Major comments:

1. No statistics are added to the figures, this needs to be fixed.
2. Fig. 2: which SAM construct was used? How many independent studies were performed? A single study with 4 mice is not ideal, even if the differences are obvious between treatment groups. It is also problematic that no controls were used in this experiment.
3. More details need to be added to the figure legends in most cases. Again, species, clearly indicate the number of immunizations, number of animals, number of independent studies, statistical analyses etc...

Minor comments:

1. Why did the authors choose Spike V2 and not Spike V8? There was a higher level of protein production from Spike V8 (Fig. 1A)?
2. What is the rationale behind introducing the F2P mutation or F6P mutation? It is not explained in the paper.
3. Fig.1: add the number of independent studies and number of animals to the legend. Did animals receive a single shot?
4. Lines 86-87: it is not clear why Ext Data Fig 7 comes after Ext Data Fig 4.
5. Methods, Western analysis: add ab dilutions to the text.

Reviewer #2 (Remarks to the Author):

The submitted paper, "A self-amplifying mRNA COVID-19 vaccine drives potent and broad immune responses at..." describes a set of experiments testing a candidate SARS-CoV-2 vaccine for immunogenicity in mice and the ability to reduce virus replication challenge of vaccinated monkeys. The candidate vaccine is a self-amplifying mRNA that encodes a prefusion-stabilized version of the spike protein. The candidate vaccine is currently being tested in clinical trials. While the studies are well designed, executed, and described, there are several limitations that need to be addressed.

Points to consider

- 1) After reading the manuscript it is apparent that the definition of protection in this study is decreased viral replication. This needs to be clearly stated throughout. Protection from severe disease cannot be assessed in NHP because they do not develop severe disease. Thus, the title of the manuscript would be more accurate if the phrase "uncontrolled SARS-CoV-2 replication" was substituted for SARS-CoV-2. Similarly, phrases like "protection from challenge" (Lines 56, 219 and elsewhere) need to be changed to acknowledge that the "protection" is limited to control of virus replication and does not extend to protection from disease.
- 2) Line 145. This statement is incorrect, for reasons detailed above, the therapeutic effect of a vaccine cannot be envaulted in an animal model that does not develop disease. There is an apparent virologic benefit from the candidate vaccine, however.
- 3) Age and gender affect immune responses/vaccine efficacy in NHPs (and mice/humans). A

breakdown of the age and gender of the animals in each study group should be included for the NHP studies as it may explain the high heterogeneity of the spike specific TH1 biased T cell responses across animals, in addition to the lack of randomizing animals based on their MAMU haplotype.

4) line 103. The authors state "While potent spike-specific immune responses were observed at all dose levels, there was no increase in serum IFN α levels following SAM immunizations at doses < 10 μ g, suggesting that innate immune pathways are not strongly activated at lower SAM doses (Extended Data Fig. 8)." This statement is true but incomplete as IFN α levels in plasma do increase following immunization with higher doses. This needs to be acknowledged in the main text rather than buried in a supplemental figure.

5) line 109-110. This statement is incorrect. PCR to detect genomic RNA and sub-genomic RNA cannot distinguish between input and replicating virus. Both genomic and sub-genomic are produced during viral replication, as genomic RNA must be packaged into daughter virions. The difference is that sgRNA PCR does not detect gRNA in virions or cells, only mRNA in infected cells.

6) lines 168-177. Comparing the results in this paper to the results of clinical trials with other mRNA vaccines is highly speculative and does not add to the paper.

7) Figure 3D. Data for frequency of IFN γ vs IL-4 secreting cells is presented, however there were no methods included for an IL-4 ELISPOT. This assay is notoriously unreliable in NHP samples. If NHP IL-4 data is included in the paper, a complete description of the assay including limits of detection and results with a positive control antigen, no stimulation and non-specific stimulation (PMA/Iono) need to be presented.

8) Figure 4. The Y axes on all graphs are inaccurately labeled "gene copies". The sgRNA PCR assay does not detect genes. It detects mRNA.

Reviewer #3 (Remarks to the Author):

The study by Rappaport et al describes the pre-clinical assessment of self-amplifying RNA vaccines in mice and non-human primates as single dose or when used in heterologous vaccination regimen. Prior studies preclinical and clinical work has described in depth the immunogenicity of these approaches in mice and humans, this study only adds incrementally to prior-work by testing efficacy in non-human primates and showing a small difference between vaccine approaches. The way data is presented (switching between readout timepoints), different numbers of mice between panels on figures, lack of statistical information on figures and within text, makes the reader question the reproducibility of the data and therefore the current state of this manuscript precludes publication at this time as an extensive re-write is required.

Major:

1) The mouse data presented jumps between different time-points post-vaccination and it is not clear whether ELISpot, ICS or ELISA have been performed on samples generated from the same vaccination experiment or has been pooled together from different experiments. To compare levels of immunogenicity between vaccine, assays (particularly ELISpot) should be performed at the same time, especially as there are differences in the level of T cells observed in animals vaccinated with the same dose and harvested at the same-timepoint between figures (ie Fig 1C SAM V2 2 weeks post-vaccination has mean SFU 12 774 vs Fig 2A 10ug SAM 2 weeks post vaccination has Mean SFU 5106). If data has been pooled from different experiments it needs to be clearly stated in the figure legend.

2) Comparing immunogenicity of vaccines in NHP to convalescent human serum is irrelevant as the dose of vaccine (relative to size) and immunogenicity is always much higher in non-human primates. All that can be concluded from the data is that all vaccination regimens induce neutralising antibodies in NHP.

The sentence referring to fold increase in antibody titres relative to convalescent human sera should be removed from the abstract.

3) It's not clear from the data (or how the data is presented) that virus is replicating in these animals as there is only a decrease in viral loads between the first post-challenge and 2nd challenge timepoint (eg D1 and D3 nasal swabs, D2 and D3 in oropharyngeal swabs). Without another measurement of vaccine efficacy (lung pathology), the only thing that can be concluded is that vaccination induces a reduction in peak viral load between vaccinated animals and controls (although an appropriate statistical test accounting for multiple comparisons needs to be performed).

4) There are no details of statistical analysis performed in mice and there are no p-values or details of statistical tests performed on data in the text, figure legends or on the figures. A clear rationale for power calculation has been performed for the NHP study but this is the only information provided. In addition, comparison between groups is performed with multiple Mann-Whitney tests without taking into account multiple comparisons.

Other points:

1) Western Blots performed with a monoclonal antibody against the S2 portion of spike (which contains the proline and furin cleavage site), how did you confirm that binding is equivalent between codon optimised proteins. Why didn't the authors use an antibody to full-length spike or S1 which is the immunodominant region, or even RBD, where there are multiple reagents available and a region which doesn't contain the furin cleavage site or proline substitutions. Sizes of proteins are different between the 2 western blots (Fig 1 and extended data fig 2B), in extended data you show full-length spike and S2 regions, Fig 1 only S2 portion of the Western blot is shown. You should show the entire blot and when describing results be clear to note that is in S2 you are detecting.

2) Immunogenicity of Adenoviral vectors in mice has been shown to be related to infectivity dose not viral particle dose (PMCID: 3396660). When comparing immunogenicity between vectors why was the vp dose used when this can lead to differences in immunogenicity. How comparable is the P:I (particle to infectivity) ratio of the different vectors.

3) Fig 1 – different number of mice between part B and part C. Is the data presented in this figure from a single vaccination experiment or pooled from different experiments, it needs a clear explanation without cherry picking data and timepoints to show the greatest effect. If data is derived from independent experiments this needs to be clearly marked up in the figure.

4) Description of results would suggest the Ch-SAM and SAM NHP challenges were performed at independent times, if so the results should be presented as separate challenges or controls marked to demonstrate comparability of infection between challenge studies.

5) Extended Figure 9 present peak subgenomic RNA between days 3 and day 10, and only for BAL and Nasal swabs. Why these timepoints (most virus is cleared already by day 3), where is the day 10 data in Figure 4 and why not present Oropharyngeal data?

6) 3ug is considered a high dose of self-amplifying mRNA in mice. McKay et al (<https://doi.org/10.1038/s41467-020-17409-9>) demonstrated antibody and T cell responses in mice with a dose as low as 0.01ug, be cautious with the use of the term low dose.

7) Authors should comment on how immunogenicity and efficacy compares relative to additional preclinical work with mRNA and Adenoviral vectors and stabilised version of spike (doi:10.1038/s41467-021-23173-1, DOI: 10.1038/s41586-020-2607-z, DOI: 10.1038/s42003-021-02443-0, DOI: 10.1038/s41586-020-2599-8).

8) Authors have habit of referring to data as potent/high levels/broad without a reference, description of results should be tempered to clearly state levels or increase relative to other groups or timepoints analysed.

9) Line 178 "Notably, our SAM vaccine demonstrates increased immunogenicity in mice compared to both the COVAC1 and ARCT-021 SAM platforms (note no published NHP data for these vaccines) (Figure 1D, Extended Data Fig. 2G, 7)" – what internal control do you have to compare immunogenicity between your study and these published results?

10) Methods reference single pool of peptides, 2 pools of peptides or 8 pools of peptides used for stimulation of cells. Why the different pools?

11) Flow-cytometry plot should also show DMSO stimulated cells to gives readers an idea of the background response

12) No fluorochromes are stated for the flow-cytometry staining panels.

13) Full details of the mouse strain should be included in the methods, ie BALBc/OldHsd or BALB/cAnNHsd

We appreciate the reviewer's constructive feedback which we have addressed point by point below. We believe that the suggested edits have improved the clarity of the manuscript, as well as strengthened the quality of the data presentation and conclusions.

REVIEWER COMMENTS

Reviewer #1 (Remarks to the Author):

Rappaport and colleagues developed and tested a self-amplifying mRNA COVID-19 vaccine in mice and NHPs. The topic is interesting, the results are nice. The manuscript is not clearly written, important information is missing, and, in general, it is hard to find the technical/experimental details in the paper. I indicated some of these issues.

Major comments:

1. No statistics are added to the figures, this needs to be fixed

Response: *We have added statistics to all figures.*

2. Fig. 2: which SAM construct was used?

Response: *The construct used was Spike-V2(F2P), this was noted in the text and has now been added to the figure legend.*

How many independent studies were performed?

Response: *Four similar repeat studies were performed for T cell analysis of this construct (a note has been added to the legend). We have added the data from one of the repeat studies to the extended data. PNA analysis was only performed for one prime/boost study for this construct. Additional studies with constructs encoding Spike-V2 or Spike-V2(F2P) encoding the Beta variant specific mutations demonstrated similar increases in T-cells and nAb titers after SAM boosting (data not shown).*

A single study with 4 mice is not ideal, even if the differences are obvious between treatment groups. It is also problematic that no controls were used in this experiment.

Response: *N = 6 mice were used for each of the four repeat studies for T cell analysis. N = 4 mice were used for the PNA assessment. Naïve samples were assessed in the PNA assay, this data has been added to the figure. Although no naïve samples were assessed in the experiment presented in Figure 2A, naïve samples were assessed in a repeat study, this data has been added to Extended data Fig. 3. Naïve samples are also included in the ICS results in Extended Figure 4.*

3. More details need to be added to the figure legends in most cases. Again, species, clearly indicate the number of immunizations, number of animals, number of independent studies, statistical analyses etc.

Response: *Additional details have been added to figure legends.*

Minor comments:

1. Why did the authors choose Spike V2 and not Spike V8? There was a higher level of protein production from Spike V8 (Fig. 1A)?

Response: *Both V2 and V8 demonstrated increased immune response in vivo compared to V1, consistent with the western data. As licensing was available for the optimization tool used for V2, but not for V8,*

and given the aggressive timeline due to the ongoing pandemic, the decision was made to move forward with V2 for business reasons.

2. What is the rationale behind introducing the F2P mutation or F6P mutation? It is not explained in the paper.

Response: Additional explanation has been added to the manuscript. The mutations were introduced to maintain the spike protein in a prefusion form which has been associated with increased expression in mammalian cells and enhanced immunogenicity which is most likely due to increased exposure of the immunogenic S1 RBD domain in an “up” rather than “down” conformation, in which the RBD is buried within the spike protein (Pallesen et al. PNAS).

3. Fig.1: add the number of independent studies and number of animals to the legend. Did animals receive a single shot?

Response: Legend revised and more details added.

4. Lines 86-87: it is not clear why Ext Data Fig 7 comes after Ext Data Fig 4.

Response: Figures have been renumbered.

5. Methods, Western analysis: add ab dilutions to the text.

Response: Dilutions of primary actin antibody 1:1,000 and anti-mouse secondary antibody 1:10,000 has been added to the text in the methods section.

Reviewer #2 (Remarks to the Author):

The submitted paper, “A self-amplifying mRNA COVID-19 vaccine drives potent and broad immune responses at...” describes a set of experiments testing a candidate SARS-CoV-2 vaccine for immunogenicity in mice and the ability to reduce virus replication challenge of vaccinated monkeys. The candidate vaccine is a self-amplifying mRNA that encodes a prefusion-stabilized version of the spike protein. The candidate vaccine is currently being tested in clinical trials. While the studies are well designed, executed, and described, there are several limitations that need to be addressed.

Points to consider

1) After reading the manuscript it is apparent that the definition of protection in this study is decreased viral replication. This needs to be clearly stated throughout. Protection from severe disease cannot be assessed in NHP because they do not develop severe disease. Thus, the title of the manuscript would be more accurate if the phrase “uncontrolled SARS-CoV-2 replication” was substituted for SARS-CoV-2. Similarly, phrases like “protection from challenge” (Lines 56, 219 and elsewhere) need to be changed to acknowledge that the “protection” is limited to control of virus replication and does not extend to protection from disease.

Response: Revisions have been made to clarify that the vaccines provide protection from viral replication post-infection. We have kept protection in the title as replication is used as a surrogate measure for protection and has been used in other publications when referring to a reduction in SARS-CoV-2 replication in NHPs (for example Mercado et al. Nature, 2020, “Single-Shot Ad26 Vaccine Protects Against SARS-CoV-2 in Rhesus Macaques”).

2) Line 145. This statement is incorrect, for reasons detailed above, the therapeutic effect of a vaccine cannot be envaulted in an animal model that does not develop disease. There is an apparent virologic benefit from the candidate vaccine, however. *Response: Statement revised.*

3) Age and gender affect immune responses/vaccine efficacy in NHPs (and mice/humans). A breakdown of the age and gender of the animals in each study group should be included for the NHP studies as it may explain the high heterogeneity of the spike specific TH1 biased T cell responses across animals, in addition to the lack of randomizing animals based on their MAMU haplotype.

Response: *Information has been added as supplemental table.*

4) line 103. The authors state “While potent spike-specific immune responses were observed at all dose levels, there was no increase in serum IFN α levels following SAM immunizations at doses < 10 μ g, suggesting that innate immune pathways are not strongly activated at lower SAM doses (Extended Data Fig. 8).” This statement is true but incomplete as IFN α levels in plasma do increase following immunization with higher doses. This needs to be acknowledged in the main text rather than buried in a supplemental figure.

Response: *Revised.*

5) line 109-110. This statement is incorrect. PCR to detect genomic RNA and sub-genomic RNA cannot distinguish between input and replicating virus. Both genomic and sub-genomic are produced during viral replication, as genomic RNA must be packaged into daughter virions. The difference is that sgRNA PCR does not detect gRNA in virions or cells, only mRNA in infected cells.

Response: *Revised for clarity and accuracy and reference added. This RT-PCR assay uses a probe in the subgenomic leader sequence to specifically assess subgenomic RNA only in infected cells (Wolfel et al. Nature, 2020) and is used as a measure of replicating virus in several published studies of vaccine efficacy in NHP (Corbett et al. NEJM, 2020, Mercado et al. Nature, 2020, van Dormalen et al. Nature, 2020).*

6) lines 168-177. Comparing the results in this paper to the results of clinical trials with other mRNA vaccines is highly speculative and does not add to the paper.

Another COVID-19 SAM vaccine, COVAC1, currently in clinical development, has been shown to induce overall weaker immune responses than the authorized mRNA vaccines, with only a 61% seroconversion rate at the 10 μ g dose¹³, compared to 100% with either the Pfizer/BioNTech mRNA vaccine at doses ranging from 10 – 30 μ g¹⁴ or the Moderna mRNA vaccine at 50 and 100 μ g¹⁵. This decreased potency may have resulted from activation of innate immune pathways that can partly restrict SAM replication and antigen expression. Another SAM vaccine platform that uses an alternative non-prefusion stabilized spike sequence and a different LNP formulation, ARCT-021, induced low nAb titers below the GMT titer of convalescent sera, demonstrating decreased immunogenicity compared to the authorized mRNA vaccines¹⁶.

Response: *This statement does not compare the results of this paper to the results of clinical trials, rather it compares the results of clinical trials with other SAM vaccines to the results of clinical trials with the currently authorized mRNA vaccines. We believe it is important to briefly review the landscape of SARS-CoV-2 self-amplifying RNA vaccines in the discussion.*

7) Figure 3D. Data for frequency of IFN γ vs IL-4 secreting cells is presented, however there were no methods included for an IL-4 ELISPOT. This assay is notoriously unreliable in NHP samples. If NHP IL-4 data is included in the paper, a complete description of the assay including limits of detection and results with a positive control antigen, no stimulation and non-specific stimulation (PMA/Iono) need to be presented.

Response: Assay description has been added to methods. A positive control stimulation was assessed in triplicate (using a pool of cells from multiple samples) on each plate for both IFN γ and IL-4. PMA/Ionomycin was used for IFN γ and SEB was used for IL-4. For both assays, the positive control stimulation generated strong responses that were too numerous to count accurately (well confluence > 35%). Therefore, this data cannot be added to the graph in Figure 3D. These details have been noted in the figure legend and the methods.

8) Figure 4. The Y axes on all graphs are inaccurately labeled “gene copies”. The sgRNA PCR assay does not detect genes. It detects mRNA.

Response: Revised.

Reviewer #3 (Remarks to the Author):

The study by Rappaport et al describes the pre-clinical assessment of self-amplifying RNA vaccines in mice and non-human primates as single dose or when used in heterologous vaccination regimen. Prior studies preclinical and clinical work has described in depth the immunogenicity of these approaches in mice and humans, this study only adds incrementally to prior-work by testing efficacy in non-human primates and showing a small difference between vaccine approaches. The way data is presented (switching between readout timepoints), different numbers of mice between panels on figures, lack of statistical information on figures and within text, makes the reader questions the reproducibility of the data and therefore the current state of this manuscript precludes publication at this time as an extensive re-write is required.

Major:

1) The mouse data presented jumps between different time-points post-vaccination and it is not clear whether ELISpot, ICS or ELISA have been performed on samples generated from the same vaccination experiment or has been pooled together from different experiments. To compare levels of immunogenicity between vaccine, assays (particularly ELISpot) should be performed at the same time, especially as there are differences in the level of T cells observed in animals vaccinated with the same dose and harvested at the same-timepoint between figures (ie Fig 1C SAM V2 2 weeks post-vaccination has mean SFU 12 774 vs Fig 2A 10ug SAM 2 weeks post vaccination has Mean SFU 5106). If data has been pooled from different experiments it needs to be clearly stated in the figure legend.

Response: Clarifying details have been added to figure legends, as well as main text. Additional data added to supplemental figures.

2) Comparing immunogenicity of vaccines in NHP to convalescent human serum is irrelevant as the dose of vaccine (relative to size) and immunogenicity is always much higher in non-human primates. All that can be concluded from the data is that all vaccination regimens induce neutralising antibodies in NHP. The sentence referring to fold increase in antibody titres relative to convalescent human sera should be removed from the abstract.

Response: As pseudovirus neutralization assays may vary widely across labs, the purpose of assessing convalescent human sera was to establish the consistency of the assays in-house and enable comparison with other published non-human primate data, as this control was included in other publications evaluating vaccine efficacy in non-human primates (Corbett et al. NEJM, 2020; Vogel et al. Nature, 2021).

3) It's not clear from the data (or how the data is presented) that virus is replicating in these animals as there is only a decrease in viral loads between the first post-challenge and 2nd challenge timepoint (eg D1 and D3 nasal swabs, D2 and D3 in oropharyngeal swabs). Without another measurement of vaccine efficacy (lung pathology), the only thing that can be concluded is that vaccination induces a reduction in peak viral load between vaccinated animals and controls (although an appropriate statistical test accounting for multiple comparisons needs to be performed).

Response: *The sgRNA RT-PCR assay uses a probe in the subgenomic leader sequence to specifically assess subgenomic RNA only in infected cells (Wolfel et al. Nature, 2020) and is used as a measure of replicating virus in several published studies of vaccine efficacy in NHP (Corbett et al. NEJM, 2020, Mercado et al. Nature, 2020, van Dormalen et al. Nature, 2020). Text has been revised to clarify conclusions.*

Analysis of peak viral load is presented in the extended data (Extended Data Fig. 9 in initial submission, now Fig. 8). Statistical analyses were performed to account for multiple comparisons (Kruskal-Wallis followed by Dunn's multiple comparison post-test), adjusted p-values are reported in the text and in Extended Data Fig. 8.

4) There are no details of statistical analysis performed in mice and there are no p-values or details of statistical tests performed on data in the text, figure legends or on the figures. A clear rationale for power calculation has been performed for the NHP study but this is the only information provided. In addition, comparison between groups is performed with multiple Mann-Whitney tests without taking into account multiple comparisons.

Response: *Statistics added to mouse data figures and legends and details added to methods. Analyses of NHP T cell and viral load data has been revised to correct for multiple comparisons, adjusted p-values are reported. Details added to text, figures, legends and methods.*

Other points:

1) Western Blots performed with a monoclonal antibody against the S2 portion of spike (which contains the proline and furin cleavage site), how did you confirm that binding is equivalent between codon optimised proteins. Why didn't the authors use an antibody to full-length spike or S1 which is the immunodominant region, or even RBD, where there are multiple reagents available and a region which doesn't contain the furin cleavage site or proline substitutions. Sizes of proteins are different between the 2 western blots (Fig 1 and extended data fig 2B), in extended data you show full-length spike and S2 regions, Fig 1 only S2 portion of the Western blot is shown. You should show the entire blot and when describing results be clear to note that is in S2 you are detecting.

Response: *At the time when we did these studies in early 2020 there were fewer SARS-CoV-2 reagents available. The anti-S1 antibody available gave high background in our assays and appeared to cross react non-specifically with ChAd antigens. The S2 antibody gave the cleanest results and was therefore selected as our antibody of choice. For the screen of the various codon optimized constructs, these constructs expressed a wild-type amino acid sequence with intact furin cleavage site and without the 2 proline substitutions. As the expressed amino acid sequence is identical between the constructs, the S2 antibody is anticipated to recognize these proteins equally well (unless they are expressed at different levels) and so we were comfortable with using the anti-S2 antibody. In the revised manuscript Figure 1A, we have included almost the entire blot with all proteins (uncleaved and cleaved intermediates) in each sample that were detected by the Western.*

For the Furin proline constructs the primary intent of the Western Blot was to confirm expression from these constructs: 2P, 6P, 2P & 6P plus Furin mutation, not to compare expression levels. The anti-S2

antibody (GeneTex 1A9) is against amino acids 1029-1192 which are outside the Furin and 2P substituted regions, and given the gel is a reducing gel we would expect this region to be similarly exposed in the Spike protein expressed from the various constructs. Others have claimed that the Furin and proline substitutions lead to higher levels of mammalian expression, and we believe that this, combined with the conformational changes, is likely leading to the enhanced immunogenicity in the extended Figure 2C-G.

2) Immunogenicity of Adenoviral vectors in mice has been shown to be related to infectivity dose not viral particle dose (PMCID: 3396660). When comparing immunogenicity between vectors why was the vp dose used when this can lead to differences in immunogenicity. How comparable is the P:I (particle to infectivity) ratio of the different vectors.

Response: This is driven by regulatory concerns about the total viral protein that a subject might be exposed to. In addition, the FDA requires dosing based on VP, as the IU assay differs between labs. We agree that the VP:IU ratio can impact immune responses if extreme differences exist between vectors, as highlighted in the referenced example: 7 versus 48 (Dicks et al. 2012). While we dose based on VP clinically (as required by the agency), we have VP:IU ratio release specifications. We have seen no more than a 2X difference in VP:IU ratio between the various preclinical vectors tested in these studies and our own studies indicate that such differences would have negligible impact on overall immunogenicity in vivo.

3) Fig 1 – different number of mice between part B and part C. Is the data presented in this figure from a single vaccination experiment or pooled from different experiments, it needs a clear explanation without cherry picking data and timepoints to show the greatest effect. If data is derived from independent experiments this needs to be clearly marked up in the figure.

Response: This has been clarified in main text and figure legends.

4) Description of results would suggest the Ch-SAM and SAM NHP challenges were performed at independent times, if so the results should be presented as separate challenges or controls marked to demonstrate comparability of infection between challenge studies.

Response: Immunizations were staggered to enable challenge at the same time. This information has been added to the methods.

5) Extended Figure 9 present peak subgenomic RNA between days 3 and day 10, and only for BAL and Nasal swabs. Why these timepoints (most virus is cleared already by day 3), where is the day 10 data in Figure 4 and why not present Oropharyngeal data?

Response: BAL was assessed at days 1, 3, 5, 7 and 10 post challenge. Nasal and oropharyngeal swabs were assessed daily for days 1 – 7 and then at day 10 and 14. Since all virus is cleared in all vaccinated animals by day 7 in BAL, day 3 in Ors, and day 5 in NS, we felt that the later timepoints do not add additional value and are not included for clarity. The complete dataset has been added as a supplemental table. To address your concern regarding viral clearance by day 3, we assume this refers to the BAL specifically, and this analysis has been revised to include days 1 – 10 for BAL (which was not assessed at day 2 or day 14). A figure has also been added for Ors (days 1 – 14). Peak sgRNA in nasal swabs is assessed between days 2 and 14, as was reported in other published studies evaluating SARS-CoV-2 vaccines in rhesus macaques (Corbett et al. NEJM, 2019; Mercado et al. Nature 2020).

6) 3ug is considered a high dose of self-amplifying mRNA in mice. McKay et al (<https://doi.org/10.1038/s41467-020-17409-9>) demonstrated antibody and T cell responses in mice with a dose as low as 0.01ug, be cautious with the use of the term low dose.

Response: We also demonstrate T cell immune response in mice at doses as low as 0.001 ug (Extended Data Fig. 5). We refer to 3 ug as a low dose in non-human primates, not mice.

7) Authors should comment on how immunogenicity and efficacy compares relative to additional preclinical work with mRNA and Adenoviral vectors and stabilised version of spike (doi:10.1038/s41467-021-23173-1, DOI: 10.1038/s41586-020-2607-z, DOI: 10.1038/s42003-021-02443-0, DOI: 10.1038/s41586-020-2599-8).

Response: Lambe et al., Spencer et al., and Mercado et al. have been referenced and discussed where applicable in the revised results section. Yang et al. describes a recombinant protein vaccine, which is outside the scope of this publication.

8) Authors have habit of referring to data as potent/high levels/broad without a reference, description of results should be tempered to clearly state levels or increase relative to other groups or timepoints analysed.

Response: Text has been revised.

9) Line 178 “Notably, our SAM vaccine demonstrates increased immunogenicity in mice compared to both the COVAC1 and ARCT-021 SAM platforms (note no published NHP data for these vaccines) (Figure 1D, Extended Data Fig. 2G, 7)” – what internal control do you have to compare immunogenicity between your study and these published results?

Response: Statement removed.

10) Methods reference single pool of peptides, 2 pools of peptides or 8 pools of peptides used for stimulation of cells. Why the different pools?

Response: In some studies, we used 8 minipools spanning Spike to assess the breadth of the T-cell response to Spike. Similarly, two peptide pools enabled measurement of S1 vs S2 specific T cell response. In other studies, we used a single Spike peptide pool due to limitations in cell number or overall size of the study. These details are noted in the legends.

11) Flow-cytometry plot should also show DMSO stimulated cells to gives readers an idea of the background response.

Response: Added to extended data Fig. 10

12) No fluorochromes are stated for the flow-cytometry staining panels.

Response: These were specified in extended data Fig. 10, as well as in the reporting summary. In the revised figure the font is increased to make these clearer in the figure, and the fluorochromes have been added to the methods.

13) Full details of the mouse strain should be included in the methods, ie BALBc/OldHsd or BALB/cAnNHsd

Response: Balb/cAnNHsd. Methods have been updated.

Peer review comments second round -

Reviewer #1 (Remarks to the Author):

The authors adequately addressed my critics.

Reviewer #2 (Remarks to the Author):

The authors have adequately addressed my concerns in the revised version of the manuscript.

Reviewer #3 (Remarks to the Author):

The authors have made extensive changes to address my comments, in particular in relation to prior lack of statistical analysis and information in figure legends. Sufficient information is now provided to readers to enable critical assessment of the data. Overall the results support the conclusions.